# Environmental Controls on Evapotranspiration and Its Components in a Qinghai Spruce Forest in the Qilian Mountains

**DOI:** 10.3390/plants13060801

**Published:** 2024-03-12

**Authors:** Guanlong Gao, Xiaoyun Guo, Qi Feng, Erwen Xu, Yulian Hao, Rongxin Wang, Wenmao Jing, Xiaofeng Ren, Simin Liu, Junxi Shi, Bo Wu, Yin Wang, Yujing Wen

**Affiliations:** 1College of Environment and Resource, Shanxi University, Taiyuan 030006, China; gaoguanlong@sxu.edu.cn (G.G.); gxy69602022@163.com (X.G.); 202123907006@email.sxu.edu.cn (Y.H.); 202223907013@email.sxu.edu.cn (J.S.); wbwubo1997@163.com (B.W.); wangyin194044@163.com (Y.W.); wenyujing1203@163.com (Y.W.); 2Academy of Water Resources Conservation Forests in Qilian Mountains of Gansu Province, Zhangye 734000, China; zywangrx@163.com (R.W.); maodanjing@126.com (W.J.); rxfksw@163.com (X.R.); 3Shanxi Laboratory for Yellow River, Taiyuan 030006, China; 4Northwest Institute of Eco-Environment and Resources, Chinese Academy of Sciences, Lanzhou 730000, China; 5Gansu Qilian Mountain Forest Eco-System of the State Research Station, Zhangye 734000, China; 6China National Forestry-Grassland Development Research Center, Beijing 100714, China; liusm_jyzx@163.com

**Keywords:** evapotranspiration, transpiration, canopy interception, boosted regression trees, environmental responses

## Abstract

Qinghai spruce forests, found in the Qilian mountains, are a typical type of water conservation forest and play an important role in regulating the regional water balance and quantifying the changes and controlling factors for evapotranspiration (ET) and its components, namely, transpiration (*T*), evaporation (*E_s_*) and canopy interceptions (*E_i_*), of the Qinghai spruce, which may provide rich information for improving water resource management. In this study, we partitioned ET based on the assumption that total ET equals the sum of *T*, *E_s_* and *E_i_*, and then we analyzed the environmental controls on ET, *T* and *E_s_*. The results show that, during the main growing seasons of the Qinghai spruce (from May to September) in the Qilian mountains, the total ET values were 353.7 and 325.1 mm in 2019 and 2020, respectively. The monthly dynamics in the daily variations in *T*/ET and *E_s_*/ET showed that *T*/ET increased until July and gradually decreased afterwards, while *E_s_*/ET showed opposite trends and was mainly controlled by the amount of precipitation. Among all the ET components, *T* always occupied the largest part, while the contribution of *E_s_* to ET was minimal. Meanwhile, *E_i_* must be considered when partitioning ET, as it accounts for a certain percentage (greater than one-third) of the total ET values. Combining Pearson’s correlation analysis and the boosted regression trees method, we concluded that net radiation (*R_n_*), soil temperature (*T_s_*) and soil water content (SWC) were the main controlling factors for ET. *T* was mainly determined by the radiation and soil hydrothermic factors (*R_n_*, photosynthetic active radiation (PAR) and *T_S_*_30_), while *E_s_* was mostly controlled by the vapor pressure deficit (VPD), atmospheric precipitation (*P_a_*), throughfall (*P_t_*) and air temperature (*T_a_*). Our study may provide further theoretical support to improve our understanding of the responses of ET and its components to surrounding environments.

## 1. Introduction

Evapotranspiration (ET) reflects the complex interactions of climate, vegetation, soil and terrain [1]. It is a key component of the Earth’s hydrological system and surface energy balance [2,3], with more than 60% of the annual global precipitation being returned into the atmosphere [4] through plant transpiration (*T*) as well as evaporation from both soil (*E_s_*) and intercepted water from wet leaves and surfaces (*E_i_*) [5]. Among these, *T* is one of the fundamental ways in which ecosystems gather water and is currently the primary water flux on Earth [6]. In addition, *T* regulates the water transport mechanism in the soil–plant–atmosphere continuum through the stomata [7] and is also essential for hydrological processes, the carbon cycle and the energy balance of ecosystems [8,9]; it is strongly influenced by plants’ physiological characteristics. For *E_s_*, it is dominated by the physical factors that result from the diffusion of water to the soil surface and the patterns of rainfall and the structural characteristics of the vegetation stand [10]. What is more, several climatological parameters like solar radiation, air temperature (*T_a_*, °C), atmospheric relative humidity (RH, %) and wind speed (*u*, m s^−1^) can, if not well measured or modelled, negatively affect the assessment of *E_s_* [11,12,13]. *E_s_* is often regarded as ineffective water consumption that causes waste or a low utilization rate of water resources [14]. However, *E_s_* can keep an ecosystem cool, further supporting vegetation photosynthesis and other functions, in some cases [15]. *E_i_* has always been neglected due to its marginal proportion in the ET of vegetation in extreme arid regions. However, for forest ecosystems with dense canopies or vegetation grown in humid environmental conditions, *E_i_* cannot be neglected, as it usually accounts for a significant proportion of ET. In such cases, the accurate partitioning of ET into *T*, *E_s_* and *E_i_* is a critical step which yields both a comprehensive insight into hydrological processes and better water management [16] and is essential to improve the modeling of land–atmosphere interactions, especially by predicting the temporal response to droughts across biomes [17,18].

There are multiple established methods for partitioning ET, ranging from simple to complex, instantaneous to continuous and point scale to satellite pixel scale. These methods were reviewed by Kool et al. [14] and classified into two categories: models and measurements. The commonly used models were the Shuttleworth–Wallace model [19] and its improved structures [20,21,22,23,24,25], the clumped model [26,27], the FAO-56 dual crop coefficient model [28,29,30] and other improved dual-source models [31,32,33], while measurements were mainly eddy covariance techniques [34,35,36,37,38] and Bowen ratio systems [39,40,41] (acquiring ET), stable isotopes [42,43,44,45] (acquiring *E_s_* or *T*), sap flow meters [46,47,48,49] (acquiring *T*), microlysimeters [50,51,52] (acquiring *E_s_*) and water collection tanks [53,54] (acquiring *E_i_*). Among these methods, the modeling approach has the advantage of its applicability over a wide range of time scales and can be applied to the spatial scale of an entire ecosystem [55,56], but these models always require complex parameterizations and still require validation. For measurements, researchers usually measure three of four parameters (ET and its components) and then calculate the last parameter based on the assumption that ET = *T* + *E_s_* + *E_i_* [10,57,58,59,60]. ET partitioning is site-specific and strongly influenced by the availability of water and energy [61], so it is still urgent to study the partitioning of ET across various vegetation types under changing climatic conditions.

As ET tends to display complex spatiotemporal patterns across scales due to the complex interactions between ET and the surrounding environments [62,63], it is of critical importance to elucidate how ET responds to a variety of environmental variables across landscapes with varying land surface and climatic conditions [64,65,66]. Though a great number of studies have explored the response characteristics of ET to different environmental factors [67,68,69,70], no unified conclusion has been reached so far [71]. For each component of ET, the dominant environmental controlling factors are also different. The environmental response of *T* has been the focus of research in the fields of water demand and supply and the adaptation mechanism of plants to their environments [72,73,74,75]. *T* is generally assumed to be affected by site hydrometeorological factors [76], such as net/shortwave radiation (*R_n_*/*R_s_*, W m^−2^), vapor pressure deficit (VPD, kPa), atmospheric precipitation (*P_a_*, mm), soil water content (SWC, %), *T_a_* and RH across temporal scales [77,78,79,80]. Among these factors, *R_n_*/*R_s_*, *T_a_* and VPD are positively correlated with *T* by controlling stomatal conductance over short timescales (i.e., hourly and daily), while for longer timescales (i.e., seasonally and interannually), SWC and *P_a_* are responsible for most of the variation observed in *T* [73,79,81]. Consequently, high *T* rates always correspond to high *T_a_*, low RH and sufficient soil water availability [82,83,84]. *E_s_*, in contrast, represents the phenomenon concerning the change in the liquid phase from water to vapor [85], which primarily depends on soil and environmental factors [16,86,87,88] and may vary between understory and bare soil circumstances. In conclusion, due to the different water consumption mechanisms, ET, *T* and *E_s_* are likely to have different responses to changes in environmental factors.

The Qilian mountains (QLMs), located in the upper reach of the Heihe River Basin (HRB), the second largest inland river basin in China, are an important ecological barrier in the northwest regions. As the constructive tree species in the QLMs, Qinghai spruce (QHS) forest is a typical type of water conservation forest and plays an important role in regulating the regional water balance. Given the complex topography and climatic conditions, the region’s QHS ecosystems are fragile and sensitive to climate change [89]. ET is the most important way to consume water for the QHS; therefore, identifying the characteristics and environmental responses of ET and its components is of great significance for the regular growth of QHS. In this study, we quantified ET, *T* and *E_i_* by direct field measurements and calculated *E_s_* based on the assumption that total ET equals the sum of *T*, *E_s_* and *E_i_*, the latter often being neglected due to its small impact on the water balance of some areas or simplifications used in calculations. Then, we analyzed the environmental controls on ET, *T* and *E_s_* by using the boosted regression trees (BRTs) method. The purposes of our study were to: (1) partition ET into *T*, *E_s_* and *E_i_* and analyze the magnitudes and changing characteristics of each parameter; and (2) identify the dominant environmental controlling factors for ET, *T* and *E_s_*.

## 2. Materials and Methods

### 2.1. Experimental Site

This study was conducted in the Pailugou watershed (38°31′–38°33′ N, 100°16′–100°18′ E), located in the middle part of the QLMs (36°43′–39°36′ N, 97°25′–103°46′ E). The watershed has an area of 2.85 km^2^ and an elevation range between 2500 and 3800 m a.s.l. In this area, the mean total daily *R_n_* is 110.28 kW m^−2^, the mean annual *T_a_* is 1.6 °C, the mean annual *P_a_* is 435.5 mm year^−1^, the mean annual RH is 60% and the mean daily *u* is between 0.1 and 2.8 m s^−1^ [90].

Our experimental site (38°33′12.15″, 100°17′8.19″, altitude 2765 m a.s.l; Figure 1) was composed of 126 pure natural QHS trees aged between 49 and 119 years. The average tree height was 10.5 m, the average diameter at breast height was 14.78 cm, the average sapwood area was 33.3 cm^2^ and the average crown breadth of the trees was 322 cm × 357 cm. The canopy density was 0.58, and the tree density was 1200 trees·ha^−1^.

### 2.2. Eddy Covariance (EC) and Meteorological Measurements

A uniform open-path EC system which consisted of a 3D sonic anemometer/thermometer (model CSAT3, Campbell Scientific Inc., Logan, UT, USA) and an open-path CO_2_/H_2_O gas analyzer (model LI-7500, Li-COR Inc., Lincoln, NE, USA) was installed in a tower to monitor CO_2_/H_2_O fluxes at a height of 24 m. The fetch of the EC system was approximately 1000 m in the predominant upwind direction. Signals were recorded by a datalogger (model CR3000, Campbell Scientific, Logan, UT, USA) and block-averaged over 30 min intervals.

Meteorological instruments were installed at heights of 2, 15 and 30 m in the tower to monitor all available meteorological variables. Among those, two sets of rain gauges (TE525MM, Campbell Scientific Inc., Logan, UT, USA) were installed at heights of 2 m and 30 m to measure *P_a_* and throughfall (*P_t_*, mm). *R_n_*/*R_s_* and photosynthetic active radiation (PAR; W m^–2^) were measured by radiometers (Li 200X, Li-COR Inc., Lincoln, NE, USA; LI-190SA, LI-COR Inc., Lincoln, NE, USA). *T_a_* and RH were measured by instruments (HMP115A, Onset, Bourne, MA, USA; HMP45A, Campbell Scientific, USA), then VPD could be calculated from *T_a_* [91]. Soil temperature (*T_s_*, °C) (109SS, Campbell Scientific, Logan, UT, USA) and SWC (SMC300, Spectrum Technologies, Plainfield, IL, USA) were measured at depths of 10, 20, 30, 40, 60 and 80 cm.

### 2.3. Sap Flow and Transpiration

Three sample trees that were representative of the surrounding stand were selected (Table 1). For each tree, a thermal dissipation probe (TDP) (RR-8210, Rainroot Ltd., Beijing, China) with two needles was inserted into the sapwood, with only the upper needle continually heated and the lower one kept normal. The sensor output signal was the difference in temperature (d*t*) between the two probes. Before installation, a small piece of bark was removed with a length between 10 and 15 cm in diameter at breast height (DBH, cm). Afterwards, two holes were drilled (the depths of which were slightly greater than 10 mm) using a drilling plate, and the probes with lengths of 10 mm were inserted into the holes. Then, the probes were covered with foam boxes and wrapped with reflective paper to prevent the influence of the natural environment on the measurement data. Eventually, the probes were connected to a datalogger and set up to collect data every 2 s, and the data were automatically averaged and stored every 10 min [92]. The sap flow velocity (*V_s_*, cm h^–1^) can be calculated according to the following formula [93]:$$V_{s}=0.0119{\left(\frac{dt_{m}-dt}{dt}\right)}^{1.231}$$
where d*t* is the difference in temperature at one moment, in °C, and d*t_m_* is the maximum difference in temperature in a day, in °C.

Then, *T* can be calculated through the following formula:$$T=\frac{0.001}{A}\sum_{i}^{n}V_{si}A_{si}N_{i}$$
where *V_si_* is the average sap flow velocity at *i* diameter class, cm h^–1^; *A_si_* is the total area at *i* diameter class, cm^2^; *n* is the total number of diameter classes; and *N_i_* is the total number of trees at *i* diameter class.

### 2.4. Stem Flow (S) and Canopy Interception

According to the distribution of the diameter class of the QHS stand, 1~3 sample trees were chosen from each diameter class to measure *S*, and a total of 10 sample trees were selected in the end. When measuring, we first made grooves by cutting several polyethylene plastic pipes and fixed them at heights of 50~140 cm above the bases of the trunks by using nails and glass cements; then, the stem flow was collected by importing it into a plastic bucket with a capacity of 10 L. *S* could then be calculated through the following formula:$$S=\sum_{i=1}^{n}\frac{C_{i}\times N_{i}}{A_{st}\times 1000}$$
where *C_i_* is the volume of *S* at *i* diameter class, mL, and *A_st_* is the area of the QHS stand, m^2^.

*E_i_* can be calculated according to the following principle of water balance:$$E_{i}=P_{a}-P_{t}-S$$

### 2.5. Flux Data Processing and Gap Fillings

CO_2_/H_2_O flux and CSAT3 data obtained from the EC system were processed using the EddyPro software version 6.0.0 (LI-COR Biosciences, Inc., Lincoln, NE, USA). Then, some corrections, including time lag compensation by covariance maximization (with default), density fluctuation compensation by the use/convert feature to the mixing ratio, block average detrending and double coordinate rotation, were made. Statistical analysis of the raw time series data was processed according to [94], and quality check flagging was obtained according to the CarboEurope standard.

The gaps in the ET data were filled by using different methods according to the gap lengths. Specifically, the gaps of less than 2 h were filled through the linear interpolation method and the gaps of less than 2 days were filled through the mean diurnal variation method [95], while the gaps longer than 2 days were filled through the energy balance equation [96].

### 2.6. Boosted Regression Trees (BRTs)

The BRTs method developed by Elith et al. [97] was used to quantify the contribution of each environmental variable to ET, *T* and *E_s_*. The BRTs method combines the strengths of regression tree algorithms and the boosting technique, which can handle different data types and accommodate missing data, and its prediction performance is superior to most traditional modeling methods in fitting complex nonlinear relationships [98]. In this study, the gbm package was used to run the BRTs model in R software (version 4.3.1); the values of the learning rate, tree complex and bag fraction were 0.01, 10 and 0.5, respectively, and the family type was Gaussian. The output results represent the relative importance of a single impact factor in the form of a percentage.

### 2.7. Statistical Analyses

The study periods we selected were from May to September (the main growing season of QHS) in 2019 and 2020, all the measured data for which were processed to hourly values. All the statistical analyses were performed using SPSS 19.0 and Origin 8.0.

## 3. Results

### 3.1. Variations in the Main Environmental Factors

Detailed information on the key meteorological factors (Figure 2) is essential to assess their variations with respect to ET and its components. During the study periods in 2019 and 2020, *R_n_*, PAR and *T_a_* showed similar trends, and the daily values of all three parameters in July and August were greater than those in other months. The PAR values were 125.9 and 124.7 Wm^−2^, which accounted for 74.8% and 73.3% of *R_n_* (Figure 2a). The average *T_a_* values were both 11.1 °C (Figure 2b). The variation trends of *T_s_* at different depths were similar, and *T_s_* showed more significant trends that increased first and then decreased compared to *T_a_* (Figure 2c). The VPD variations were rather consistent with *T_a_*, as VPD was calculated from the equation based on *T_a_*. The average VPD values were 0.57 and 0.61 kPa and ranged from 0.02 to 1.38 and 0.03 to 1.44 kPa, respectively (Figure 2b). *P_t_* accounted for 76.8% and 89.7% of *P_a_*, indicating that a significant proportion of precipitation had been intercepted (Figure 2d). The variation trends in SWC were consistent with the amounts of *P_t_* and *P_a_*, and this was because precipitation is the only source of water input for the QHS ecosystem (Figure 2e). The average values for RH were 62.8% and 60.6%, and they ranged from 23.8% to 97.1% and 16.1% to 96.7%, respectively. *u* remained relatively stable (there were no significant seasonal differences), and its variation was fairly similar each year (its average values were 0.58 and 0.59 m s^−1^ and ranged from 0.05 to 0.99 and 0.03 to 0.93 m s^−1^, respectively) (Figure 2f).

### 3.2. Variations in ET, T and E_s_

Monthly mean diurnal variations in ET and *T* are shown in Figure 3. Both parameters increased rapidly from sunrise and reached a maximum at approximately midday and then decreased (Figure 3a,b), and the maximum values of *T* were greater than 0.3 mm h^−1^. We can also see the much lower values of *T* between sunset and sunrise on the next day (Figure 3b), which confirmed the existence of nocturnal transpiration. *E_s_* and *E_i_* values were not analyzed at the monthly mean diurnal scale, as *E_i_* was not measured hourly and *E_s_* was calculated by the difference between ET and *T* and *E_i_*.

The monthly dynamics of daily variations in ET, *T* and *E_s_* during the main growing season of the QHS are shown in Figure 4. The peak values for ET (4.4 and 4.3 mm) and *T* (2.8 and 2.7 mm) occurred at the end of June or in July. Both parameters increased first and then decreased during the whole study period in each year, especially for *T*, which showed even more significant trends, as the period (between July and August) was the flourishing growing stage of the QHS (Figure 4a,b). Total values between July and August were 108.3 and 104.0 mm for *T*, which accounted for 57.2% and 56.0% of the total amounts in 2019 and 2020, respectively. There were no similar variation trends for *E_s_* compared to ET and *T* (Figure 4c). This is because *E_s_* was mainly controlled by the hydrothermic condition of the soil and was not correlated with plant growth.

Monthly cumulative values for ET, *T* and *E_s_* during the study periods in each year are shown in Figure 5. The total amounts of ET were 353.7 mm in 2019 and 325.1 mm in 2020, and, among all the components, *T* was always the largest part (189.5 mm in 2019 and 185.7 mm in 2020). ET values in each month in both years were not significantly different, except for July, during which the difference was 24.6 mm (Figure 5a), and this was mainly due to the significant difference in *P* (51.1 mm) during the period (Figure 2d), which influenced *E_s_* rather than *T* (Figure 5b,c).

### 3.3. Variations in T/ET, E_s_/ET and E_i_/ET

Monthly dynamics of daily variations in *T*/ET and *E_s_*/ET are shown in Figure 6. *T*/ET increased until July and gradually decreased afterwards, while *E_s_*/ET showed opposite trends. The daily maximum *T*/ET and *E_s_*/ET values can both reach up to as high as 90%. In each month, in both years, *T*/ET was nearly always the greatest among all the components, except for June in 2019, when the *T*/ET (46.8%) value was slightly lower than that for *E_i_*/ET (48.5%), and this was mainly due to the significant amount of *P* (104.6 mm) in the period. Overall, *T* accounted for at least one-third of the ET and *E_s_*/ET accounted for as little as 3%, while *E_i_*/ET was uncertain, as it was mainly controlled by the amount of *P* (Figure 7). From the total amounts of *T*, *E_s_* and *E_i_*, the *T*/ET values were always the greatest at 53.6% in 2019 and 57.1% in 2020. *E_i_* accounted for a certain percentage of ET (32.9% in 2019 and 31.6% in 2020), which indicated the important role of *E_i_* in the water balance of the Qinghai spruce forest ecosystem. The contribution of *E_s_* to ET was minimal among all the components, which reflects the excellent water conservation function of the QHS ecosystem (Figure 8).

### 3.4. Dominant Environmental Controlling Factors for ET, T and E_s_

Firstly, we selected the environmental factors that were significantly related to ET, *T* and *E_s_* by using Pearson’s correlation analysis. Then, we used the BRTs model to identify the environmental factors that greatly contributed to ET, *T* and *E_s_*. Pearson’s correlation analysis revealed that ET was significantly positively correlated with *R_n_* and PAR (the correlation coefficients were 0.75 and 0.76, respectively) and then *u*_2_ and *u*_15_ (the correlation coefficients were 0.51 and 0.57, respectively). *T* was significantly positively correlated with *T_a_*, *R_n_*, PAR, *T_s_* and SWC, with correlation coefficients all greater than 0.5, and *E_s_* was significantly positively correlated with VPD (for which the correlation coefficient was 0.51). In addition, ET, *T* and *E_s_* were all negatively correlated with RH (Figure 9).

Then, we conducted a multicollinearity assessment of the environmental variables and selected those which passed the multicollinearity test (the absolute value of the related coefficient was lower than 0.8, and the variance inflation factor was lower than 10). Eventually, some of the many environmental factors were selected as the predictor variables. The BRTs analysis showed that, among the selected predictor variables for ET, *T* and *E_s_*, ET was mainly controlled by *R_n_*, *T_s_*_30_ and SWC_40_, with contributions at 51.0%, 15.7% and 8.5%, respectively. *T* was mainly controlled by *R_n_* and PAR, with contributions at 34.9% and 34.2%, respectively, and then *T_s_*_30_ (the contribution was 13.2%). For *E_s_*, it was mainly controlled by VPD, *P_t_*, *T_a_* and *P_a_*, with contributions at 28.5%, 11.8%, 10.9% and 9.9%, respectively (Figure 10).

## 4. Discussion

### 4.1. Partitioning Methods for ET

Accurately partitioning ET into *T*, *E_s_* and *E_i_* can not only help us better understand the water and energy exchanges between the surface and atmosphere, but also determine the water demand and further improve water resource management [15]. Thus, partitioning ET is crucial for understanding water resources in the context of global climate change [99]. ET can be partitioned mainly through the methods of modeling and measuring. For modeling, the SW model and the clumped model and their improved forms have been widely used due to their suitable physical mechanisms. Researchers nowadays mainly focus on optimizing the structures of the dual- or multi-source models [20,22,100,101,102], but these models incorporate a large number of parameters, some of which are difficult to parameterize [31]. The FAO-56 dual crop coefficient model is an indirect method for calculating ET from reference crop ET and crop coefficients. Based on the basal crop coefficient and the soil evaporation coefficient recommended by the FAO, the FAO-56 dual crop coefficient model has been widely used in estimating crop ET and its components. However, crop coefficients for many types of vegetations have not been provided by the FAO. In general, though great efforts have been made and modeling accuracies have also been improved, there are limitations to the use of these models. Moreover, modeling can never represent the true amounts of ET and its components, and the applicability of the models for different ecosystems still requires extensive verification. In contrast, measuring methods can accurately quantify ET and its components. The commonly used measuring methods include stable isotope [103,104,105,106], sap flow and microlysimeters [10,22,107,108]; the EC high-frequency correlation approach [109,110,111,112]; and large-aperture scintillometers with Bowen ratio systems [113,114,115,116]. Studies based on these instruments and the assumption that ET equals the sum of all the components have mostly focused on partitioning ET into *T* and *E_s_* [60,117,118,119,120,121,122,123]; in fact, *E_i_* should also be considered when it accounts for a significant percentage.

### 4.2. Proportions of T and E_i_ on ET

Understanding the annual variation in *T*/ET remains a challenge and is essential to garner a thorough understanding of plant responses to the changing environment [124]. A number of studies carried out with different methodologies have identified that there exists a large variability (appropriately 20–90%) in *T*/ET across biomes or even at the global scale [125]. For a typical inland river basin, among the various environmental conditions, the *T*/ET for vegetations in arid and semi-arid climate zones in lower reaches with sufficient water supply was between 20 and 70% [126,127,128,129], while for vegetations in high-altitude mountains in upper reaches, the mean values of this ratio were slightly greater. For the QHS in the QLMs in the upper reaches of the Heihe River in our study, the *T*/ET values were 53.6% and 57.1% in 2019 and 2020, respectively, which were close to the results for the QHS in the QLMs obtained by other researchers [130,131], while being lower than those obtained for subtropical coniferous plantations, e.g., 63~68% for Wei et al. [132], 69~85% for Zhu et al. [133] and 77.4% for Ren et al. [134]. Monthly variations in *T*/ET showed that *T* contributed the most to the ET, especially in summer (the *T*/ET reached up to 75% on July in 2020), indicating that *T* is the most important factor for ET consumption and that it changes with the growing stages of the QHS.

*E_i_* is an essential component of the forest hydrological cycles. Studies on *E_i_* processes have been conducted on various plants, all of which clearly indicated that it cannot be ignored as an important component of water balance [130,135,136,137]. In our study, the *E_i_*/ET values for the QHS in the QLMs were 32.9% and 31.6% in 2019 and 2020, respectively, which further emphasized that *E_i_* plays an undeniable role in ET. The amount and percentage of *E_i_* vary between climatic regions [138], as they depend on factors such as vegetation characteristics [139,140], meteorological conditions [141] and precipitation characteristics [142,143]. According to previous studies of the QHS in the QLMs, *E_i_* is mainly controlled by *P*. Generally speaking, a low amount or intensity of *P* leads to small impacts on the branches and leaves, and the rainwater can be intercepted with great probability. Furthermore, the high values for the leaf area index (the maximum value was 3.96) of the QHS in the QLMs also contributed to the greater *E_i_*/ET.

Proportions of *T*, *E_s_* and *E_i_* on the ET of the QHS and other coniferous forests are summarized in Table 2.

### 4.3. Environmental Controls for ET, T and E_s_

The role of environmental variables in controlling ET and its components is an important but not well-understood aspect in arid areas in northwestern China. In our study, we concluded that *R_n_*, *T_s_* and SWC were the main controlling factors for ET, and numerous previous studies reached the same conclusions [150,151,152,153,154,155,156]. Among these, *R_n_* is the primary driver of ET, as radiation is the energy source and most of the radiant energy absorbed by the leaves is used for *T* [157]. SWC can not only affect the development of the leaf area index, but also directly and positively affect surface conductance by decreasing soil surface resistance and stomatal resistance and then increasing soil evaporation and transpiration [146]. The effect of *T_s_* on ET can be explained, as rising *T_s_* can accelerate processes of *E_s_* and *T* directly by promoting the roots’ absorption of water from the soil, and the water conservation function of the QHS ensures the presence of an adequate water resource in the soil. For *T* and *E_s_*, the main environmental controlling factors were different. *T* was mainly determined by the radiation and soil hydrothermic factors (*R_n_*, PAR and *T_S_*), which was consistent with the work by Du et al. [158] and Yang et al. [152], who also reached the same conclusion for the QHS in the QLMs, and also consistent with the work by Chen et al. [159,160]. VPD contributed less than the radiation and soil hydrothermic factors in our study, which was also consistent with the work by Du et al. [158], and the reason for this is that VPD is not the most important controlling factor, but the result of the comprehensive effect of meteorological factors to some extent [158]. *E_s_* was mostly controlled by VPD, *P_a_*, *P_t_* and *T_a_*, which was consistent with the work by Wang et al. [158], who also reached the same conclusion for the QHS in the QLMs. The climate of the QLMs is typically continental, and in this area, especially after spring, the Mongolia high press reduces and the subtropical high pressure expands gradually, which results in *T_a_* and *u* increasing rapidly, thus promoting the *E_s_* process. Consequently, it is of great significance to keep the soil moisture needed for the regular growth of the QHS and prevent spring drought in the QLMs [161].

## 5. Conclusions

Identifying ET partitioning results, as well as the variations and controlling factors for ET and its components, is essential for ensuring the regular growth of the QHS in the QLMs. In this study, we analyzed all these aspects, and the primary results are as follows: (1) ET increased first and then decreased during the main growing season of the QHS, with total values of 353.7 and 325.1 mm in 2019 and 2020, respectively. (2) The monthly dynamics of the daily variations in *T*/ET and *E_s_*/ET showed that *T*/ET increased until July and gradually decreased afterwards, while *E_s_*/ET showed opposite trends. (3) *T* always accounted for the largest part of ET and was significantly greater than *E_s_*; meanwhile, *E_i_* cannot be neglected, as it accounts for a certain percentage. (4) The main environmental controlling factors for ET and its components were different. ET was mainly controlled by *R_n_*, *T_s_* and SWC; *T* was mainly determined by the radiation and soil hydrothermic factors (*R_n_*, PAR and *T_S_*_30_); and *E_s_* was mostly affected by VPD, *P_a_*, *P_t_* and *T_a_*.

## Figures and Tables

**Figure 1 plants-13-00801-f001:**
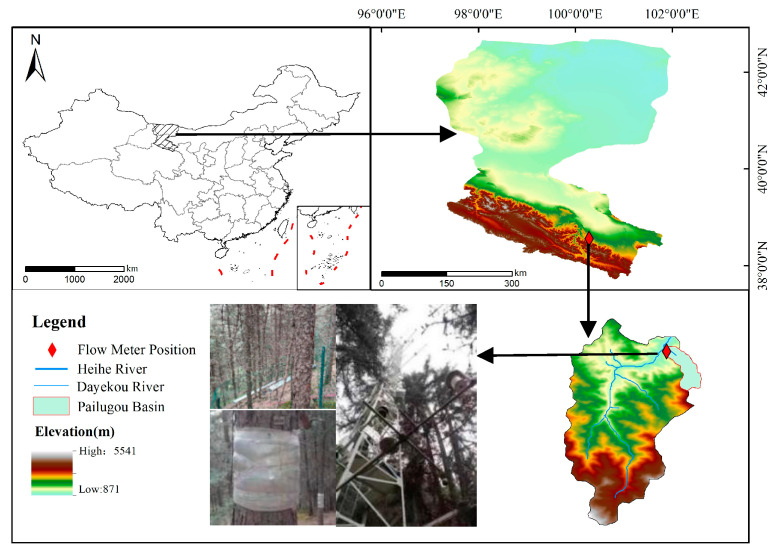
Location of the study area.

**Figure 2 plants-13-00801-f002:**
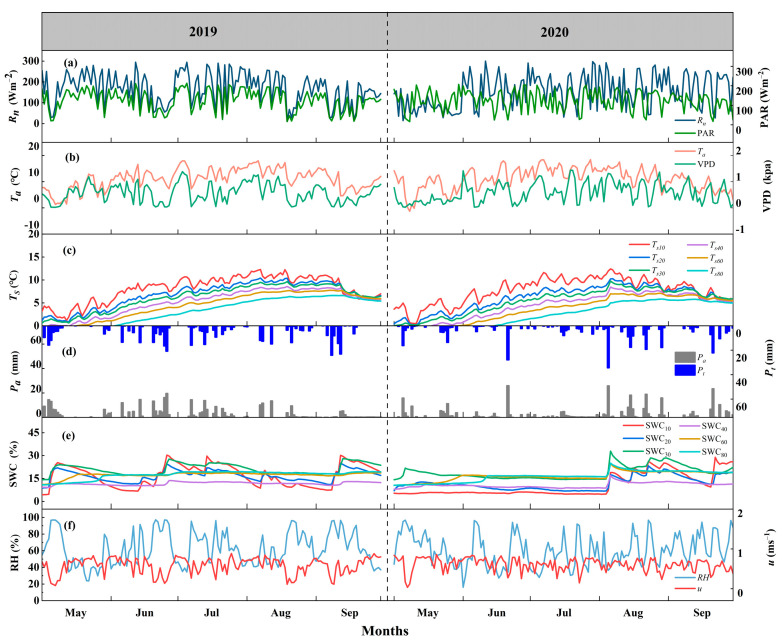
Diurnal variations in environmental factors, including (**a**) net radiation (*R_n_*) and photosynthetic active radiation (PAR), (**b**) air temperature (*T_a_*) and vapor pressure deficit (VPD), (**c**) soil temperature (*T_s_*) at different depths, (**d**) precipitation (*P_a_*) and throughfall (*P_t_*), (**e**) soil water content (SWC) at different depths, and (**f**) relative humidity (RH) and wind speed (*u*), in 2019 and 2020 at the study site.

**Figure 3 plants-13-00801-f003:**
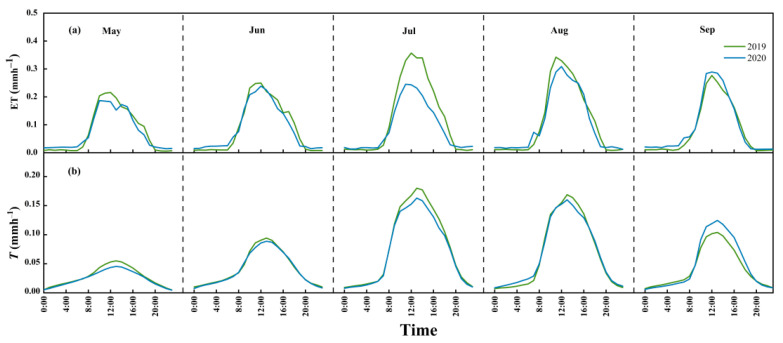
The dynamics of monthly mean diurnal variations in (**a**) evapotranspiration (ET) and (**b**) transpiration (*T*) of the Qinghai spruce in the Qilian mountains in 2019 and 2020.

**Figure 4 plants-13-00801-f004:**
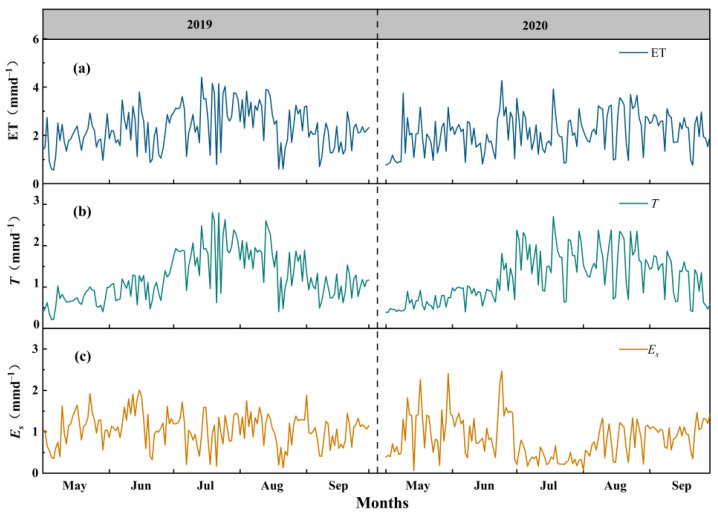
The monthly dynamics of daily variations in (**a**) evapotranspiration (ET), (**b**) transpiration (*T*) and (**c**) evaporation (*E_s_*) during the main growing season of the Qinghai spruce in the Qilian mountains in 2019 and 2020.

**Figure 5 plants-13-00801-f005:**
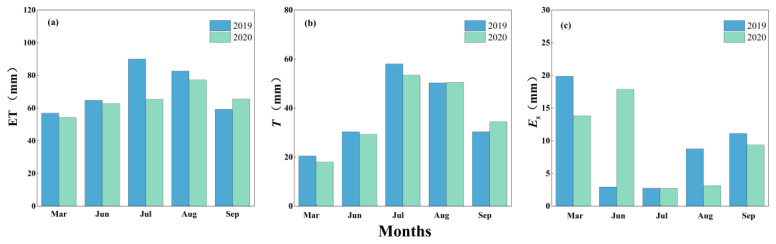
Monthly cumulative values of (**a**) evapotranspiration (ET), (**b**) transpiration (*T*) and (**c**) evaporation (*E_s_*) for the Qinghai spruce in the Qilian mountains during the study periods in 2019 and 2020.

**Figure 6 plants-13-00801-f006:**
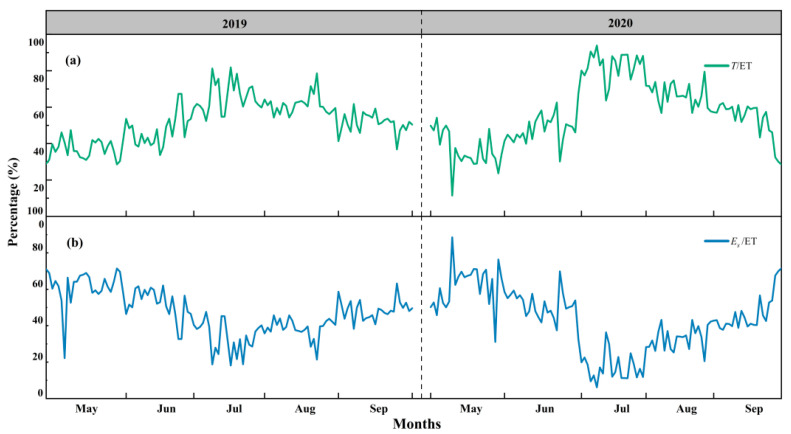
Monthly dynamics of daily variations in the proportions of (**a**) transpiration to evapotranspiration (*T*/ET) and (**b**) evaporation to evapotranspiration (*E_s_*/ET) of the Qinghai spruce in the Qilian mountains in 2019 and 2020.

**Figure 7 plants-13-00801-f007:**
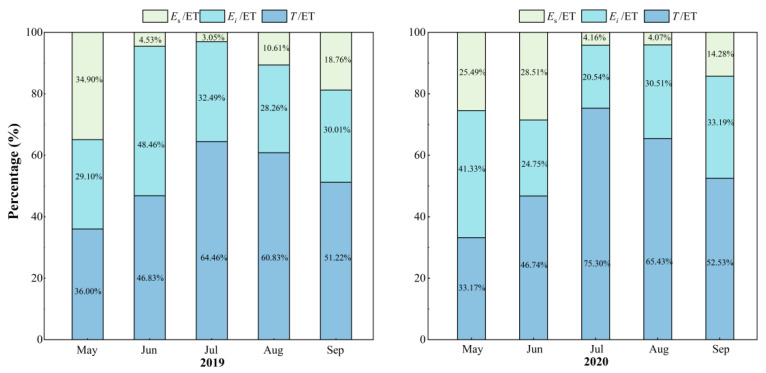
Proportions of transpiration to evapotranspiration (*T*/ET), evaporation to evapotranspiration (*E_s_*/ET) and canopy interception to evapotranspiration (*E_i_*/ET) of the Qinghai spruce in each month in 2019 and 2020.

**Figure 8 plants-13-00801-f008:**
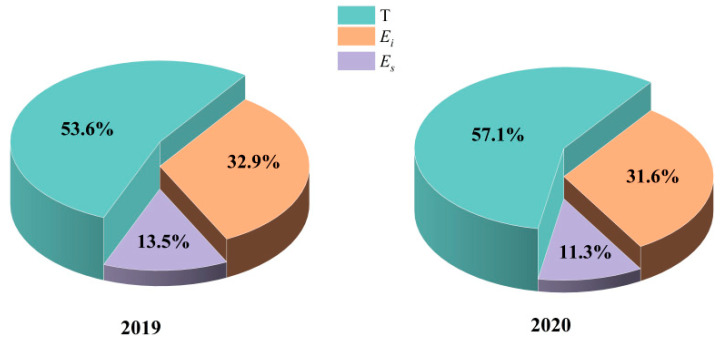
Proportions of transpiration to evapotranspiration (*T*/ET), evaporation to evapotranspiration (*E_s_*/ET) and canopy interception to evapotranspiration (*E_i_*/ET) of the Qinghai spruce during the study periods in 2019 and 2020.

**Figure 9 plants-13-00801-f009:**
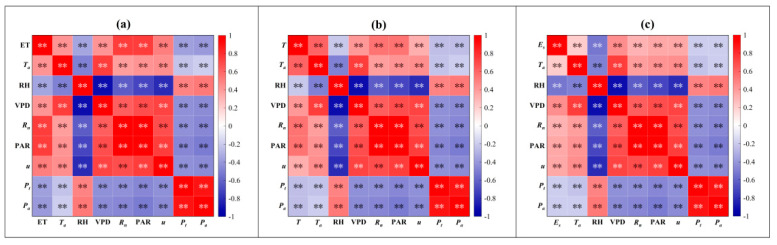
Pearson’s correlation analysis between (**a**) evapotranspiration (ET), (**b**) transpiration (*T*), (**c**) evaporation (*E_s_*) and the environmental factors, including air temperature (*T_a_*), relative humidity (RH), vapor pressure deficit (VPD), net radiation (*R_n_*), photosynthetic active radiation (PAR), wind speed (*u*), precipitation (*P_a_*) and throughfall (*P_t_*). ** means significant correlation.

**Figure 10 plants-13-00801-f010:**
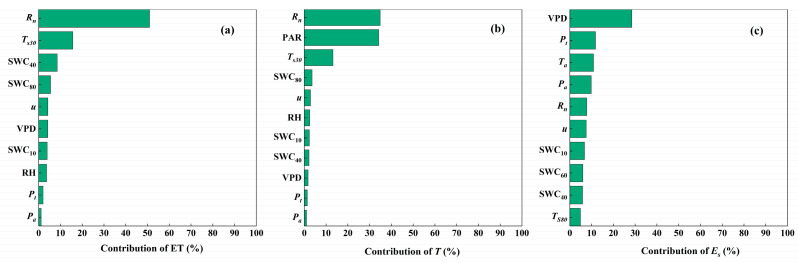
Contributions of environmental factors to (**a**) evapotranspiration (ET), (**b**) transpiration (*T*) and (**c**) evaporation (*E_s_*) based on the boosted regression trees method. The environmental factors included net radiation (*R_n_*); soil temperature at depths of 30 cm (*T_s_*_30_) and 80 cm (*T_s_*_80_); soil water content at depths of 10 cm (SWC_10_), 40 cm (SWC_40_), 60 cm (SWC_60_) and 80 cm (SWC_80_); vapor pressure deficit (VPD); relative humidity (RH); wind speed (*u*); precipitation (*P_a_*); and throughfall (*P_t_*).

**Table 1 plants-13-00801-t001:** Summary of biological parameters for the three selected Qinghai spruce trees.

Tree Number	Height (m)	DBH (cm)	*A_s_* (cm^2^)
1	11.0	14.9	111.1
2	13.5	21.0	186.5
3	15.5	37.4	525.7
Sig. (two-tailed)	0.331	0.736	0.912

Significant differences were tested by the Student’s *t*-test at a significance level of *p* = 0.05. DBH, diameter at breast height; *A_s_*, sapwood area.

**Table 2 plants-13-00801-t002:** Summary of transpiration (*T*), evaporation (*E_s_*) and canopy interception (*E_i_*) on evapotranspiration (ET) of Qinghai spruce and other coniferous forests.

Tree Species	Coordinates	Altitude (m)	Study Periods	*T*/ET	*E_s_*/ET	*E_i_*/ET	References
Qinghai spruce	38°31′–38°33′ N 100°16′–100°18′ E	2700	2019–2020	53.6% 57.1%	32.9% 31.6%	13.5% 11.3%	This study
Qinghai spruce	38°32′ N, 100°15′ E	2835	2008	51.3%	16.5%	32.2%	Tian et al. [127]
Qinghai spruce	38°31′–38°33′ N 100°17′–100°18′ E	2700 2900	2011	52.2–88.4%	11.6–47.8%		Peng et al. [131]
Qinghai spruce		2605	2001–2002	60.9%	1.1%	38.0%	Dong [144]
Qinghai spruce	36°43′–37°23′ N 100°51′–101°56′ E	2519	2019	82.6%	17.4%		Wang [145]
Coniferous and broad-leaved mixed forest	42°24′ N, 128°05′ E	738	2003–2008	65.7%	19.3%	15.0%	Lu et al. [146]
Evergreen coniferous forest	26°44′ N, 115°03′ E	102	2003–2008	61.4%	10.8%	27.8%	
Artificial coniferous forests	26°44’ N, 115°03’ E	110.8	2003–2008	65.0%	23.0%	12.0%	Wei et al. [132]
Larch forest	35°15′–35°41′ N 106°09′–106°30′ E	2264	2012	41.6%	27.2%	31.19%	Cao et al. [147]
Pine forest	35°15′–35°41′ N 106°09′–106°30′ E	2264	2012	47.2%	17.2%	35.5%	
*Pinus tabuliformis* forest	35°15′–35°41′ N 106°09′–106°30′ E	2264	2012	42.4%	27.1%	30.5%	
*Pinus yunnanensis* forest	27°00′ N, 100°10′ E	3250	2019	59.0–81.0%			Han et al. [148]
Coniferous broad-leaved mixed forest	29°20′–30°20′ N 101°30′–102°15′ E	7556	2016	48.0%			Sun et al. [149]
Evergreen coniferous forest	29°20′–30°20′ N 101°30′–102°15′ E	7556	2016	50.0%			

## Data Availability

Data are contained within the article.

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
