# Peer review of "Environmental Controls on Evapotranspiration and Its Components in a Qinghai Spruce Forest in the Qilian Mountains"

_plants, 2024, doi:10.3390/plants13060801_

Round 1

Reviewer 1 Report

Comments and Suggestions for Authors

Reviewer 2 Report

Comments and Suggestions for Authors

The manuscript submitted for review fits into the important topic of the environmental water cycle and its impact on climate and ecosystems. The authors analysed and evaluated the influence of environmental factors on the magnitude of the evapotranspiration process of a water-conserving forest in the Qilian Mountains, China, including key meteorological and soil elements (temperature and soil moisture). The core of the approach is the division of the water dispersion process into evaporation, transpiration and interception - the latter often neglected in analyses due to measurement difficulties, especially the lack of standard measurement methods for this factor, the small impact on the water balance of some areas, or simplifications used in calculations. The analyses were based on the results of direct field measurements carried out in 2019-2020, including advanced measurement methods such as eddy covariance. In addition to obtaining information on meteorological parameters, the authors carried out direct measurements of sap flow, which allowed the determination of transpiration, and stem flow, which allowed the calculation of interception. This methodological approach allowed them to fully achieve the stated objective of the manuscript.

The paper begins with an introduction to the issues analysed. It concludes with the clearly given purpose. The methodology and calculation methods are very clearly presented. The chapter on results and discussion is also very good and enriched with clear graphics. The whole work is summarised with synthetic conclusions.

In my opinion, the reviewed manuscript is a very valuable scientific study. My only objection is the lack of characterisation of the weather conditions in relation to the climatological norm for the area analysed. It is true that the authors give normative values in lines 123-124 and literally describe the course of the values of the meteorological parameters in subsection 3.1, but for greater precision I propose to classify these conditions (thermal and precipitation) whether they correspond to the norm. This is important from the point of view of the interpretation of the test results obtained - because only in relation to specific weather conditions should these results be referred to.

I recommend, taking into account the reviewer's comment, to accept the paper for publication in the journal Plants.
